# Neurotoxicity of Lanthanum Salts: A Narrative Review of Mechanistic Insights from Cellular and Animal Models

**DOI:** 10.3390/molecules30183748

**Published:** 2025-09-15

**Authors:** Tudor Mihai Magdas, Constantin Bodolea, Claudia Gherman, Ariana Raluca Hategan, Dana Alina Magdas, Maria David, Roxana Denisa Capras, Gabriela Adriana Filip

**Affiliations:** 1Department of Anatomy and Embriology, “Iuliu Hațieganu” University of Medicine and Pharmacy, 8 Victor Babeş Street, 400012 Cluj-Napoca, Romania; tmmagdas@gmail.com (T.M.M.); capras.roxana@umfcluj.ro (R.D.C.); gabriela.filip@umfcluj.ro (G.A.F.); 2National Institute for Research and Development of Isotopic and Molecular Technologies, 67-103 Donat Street, 400293 Cluj-Napoca, Romania; ariana.hategan@itim-cj.ro (A.R.H.); alina.magdas@itim-cj.ro (D.A.M.); maria.david@itim-cj.ro (M.D.); 3Department of Surgery–Practical Abilities, “Iuliu Hațieganu” University of Medicine and Pharmacy, 400337 Cluj-Napoca, Romania; gherman.claudia@umfcluj.ro; 4Anesthesia and Intensive Care Department, “Iuliu Hațieganu” University of Medicine and Pharmacy, 400347 Cluj-Napoca, Romania

**Keywords:** rare earth elements, lanthanum, toxicity, neurotoxicity, environmental pollutants

## Abstract

Driven by rapid technological development, the rising need for rare earth elements (REEs) has led to increased extraction, processing, and utilization of these materials in new technologies. Nevertheless, the waste produced by these processes is often inadequately managed, resulting in the release of these elements into the environment, contaminating water sources and the food chain, and ultimately impacting both humans and animals. Recent studies have underlined the toxic potential of these elements, raising major concerns regarding human health. Given the exponential increase in the use of this group of elements, it is essential to acquire a deep understanding of the adverse consequences associated with REEs. This review aims to examine the neurotoxic effects of lanthanum, an element of this group that has garnered significant attention from the scientific community, by providing a comprehensive overview of the mechanisms underlying its toxicity.

## 1. Introduction

Rare earth elements (REEs) refer to a group of 17 elements, encompassing 15 lanthanides along with scandium (Sc) and yttrium (Y), as defined by the International Union of Pure and Applied Chemistry. REEs can also be divided based on their atomic mass into light rare earth elements (LREEs), including lanthanum (La), cerium (Ce), praseodymium (Pr), neodymium (Nd), promethium (Pm), samarium (Sm), and europium (Eu), and heavy rare earth elements (HREEs), represented by gadolinium (Gd), terbium (Tb), dysprosium (Dy), holmium (Ho), erbium (Er), thulium (Tm), ytterbium (Yb), lutetium (Lu), and yttrium (Y). Despite their name, the natural abundance of these elements in the Earth’s crust is relatively high, even higher than that of silver or mercury. The widespread and growing relevance of REEs can be attributed to their unique magnetic, phosphorescent, and catalytic properties.

These properties have made them indispensable in a vast array of modern technologies [1,2]. In the industrial sector, they are integrated into electronics, high-capacity batteries for mobile phones and electric vehicles, LED lighting, superconductors, and solar panels. In agriculture, they are used as fertilizers and new feed additives. These elements are significant in medical technologies, with applications in both diagnostics—where they are incorporated into X-ray machines and used as MRI contrast agents—and therapeutics, where lanthanum compounds are used as phosphate binders in chronic kidney disease and lanthanide complexes as anti-cancer agents [3,4]. Current evidence suggests that gadolinium can be retained in the brain at low levels following repeated or high-dose gadolinium-based contrast agent exposure, with the extent and mechanisms of retention being influenced by patient pathology, renal function, and the type of contrast agent used; however, the clinical significance of this retention is still uncertain [5].

Because of their importance in various high-interest domains, the global demand for REEs is continuously increasing, leading to an expansion of their production outside of China, which currently dominates the market [6]. In this regard, alternative resources or recycling methods are being targeted to ensure a minimum of uncontrolled release of these elements into the environment [7,8]. This is because, despite being generally found in small quantities within water, plants, the atmosphere, or soil, there is increasing concern regarding REE accumulation in these environmental media due to anthropogenic activities and their possible toxicological effects [9].

Exposure to REEs needs to be carefully followed, monitored, and examined along the entire life cycle, from mining and extraction, through their various paths of utilization, and, ultimately, by their uncontrolled and unregulated release into the environment as waste. Recent studies conducted in China have shown that REE contamination through exposure to e-waste sites is strongly correlated with increased concentrations of the thyroid-stimulating hormone (TSH) and higher oxidative stress [10]. Despite this, the main exposure to REEs is in countries (i.e., China) that cover a wide spectrum of activities directly related to this specific element group, from mining and exploitation to the end of their life cycle. However, due to the global use and disposal of electronic devices, the harmful effects caused by these elements are no longer present only in countries such as China but have become widespread. Nowadays, human exposure is caused by the presence of these elements in air, soil, and water (Figure 1) [2]. This is because the good mobility of LREE ions facilitates their propagation and absorption from soil to plants, which subsequently represents the main entry point into the food chain (Figure 2) [11,12,13,14]. This fact leads to a strong correlation between the REE content in food and the soil composition from a specific geographical area, making the LREE content a powerful geographical indicator. This observation was demonstrated in our previously reported studies, which emphasized that there is a direct correlation between the REE content in food and its place of production [15,16].

All these exposure pillars raise concerns for human health, as recent studies have shown the negative effects of these elements [17,18,19]. In this context, the presence of lanthanides in the environment was demonstrated to negatively influence the cognitive functions of children (reduced learning and attention capacity and lower intelligence quotient (IQ) scores) [20].

Despite the overall toxicological effects of REEs mentioned in the literature, the greatest attention has been given to lanthanum and cerium because of their higher mobility, which facilitates their transfer from soil to plants and further along the entire food chain to animals and humans. Moreover, the study conducted by Briner et al. [21] stated that La encountered through water consumption can act as a behavioral teratogen. Although there is some evidence suggesting neurotoxic effects associated with other REEs, such as neodymium, gadolinium, and yttrium, research on these elements remains limited at present [22,23,24].

Based on the available data from the literature, the studies that analyze the effect of these REEs on human health report the use of lanthanum salts—lanthanum(III) chloride hydrate (LaCl_3_ xH_2_O) and lanthanum(III) nitrate hexahydrate—due to their greater hydrosolubility compared to their corresponding carbonates, which facilitates their absorption in the gastrointestinal tract [25,26]. This property raises concerns about their toxicity because, once absorbed, such elements can accumulate in various tissues, including the liver and bones, and have been shown to cross the blood–brain barrier [27,28]. This accumulation occurs directly within the brain, where deleterious effects of exposure have been reported in numerous experimental animal studies and hinted at in humans, as evidenced by REE accumulation within normal and cancerous brain tissue [29]. Studies have quantified increased REE concentrations following environmental exposure to lanthanum, while in vitro studies of lanthanum salt exposure of murine cerebral endothelial cells (bEnd.3), the first line of defense in the central nervous system, have shown decreased viability and apoptosis of cells. These results suggest the toxicity of LaCl_3_ for the BBB, causing morphological disruptions [28]. Regarding the accumulation of lanthanides in various cerebral structures, available data suggest that the accumulation might be uneven among those structures, with one study reporting a concentration in the cerebral cortex that was higher compared to the hippocampus or the cerebellum [30], while other studies found similar concentrations across these structures [31].

Besides the above-mentioned effects, reported environmental exposure studies establish that REEs can also cross the placental barrier, potentially increasing neural tube defect (NTD) risk in developing embryos; however, findings remain inconclusive, as other research reports no significant association [32,33].

This review provides a comprehensive overview of the neurotoxicity of lanthanum (La), the most commonly studied rare earth element (REE) [18]. While previous works have broadly addressed REE toxicity, our scope is specifically focused on the neurotoxic effects of La following experimental exposure. We offer a granular analysis of its underlying molecular mechanisms by integrating findings from both in vivo animal models and in vitro cellular studies to build a comprehensive mechanistic picture. By critically synthesizing the literature and highlighting conflicting data, this review seeks to enhance the current understanding of La’s neurotoxicity and identify key gaps for future research.

## 2. Results

A total of 27 studies were included, of which 8 were conducted using cellular models, including primary cultured astrocytes and neurons from Wistar rats, BV2 microglia from Kunming mice, and murine cerebral endothelial cells (bEnd.3). The remaining 19 studies involved animal models, including Wistar rats, Sprague Dawley rats, CD-1 (ICR) mice, and Chinese Kunming mice. Among the animal studies, 5 focused on adult rats (primarily Wistar), examining chronic or postweaning exposure, 13 used pregnant Wistar rats and/or their offspring, with exposure occurring via placental transfer, lactation, and postweaning drinking water, and 1 study used the nematode *Caenorhabditis elegans*. All the included studies were conducted in China. The key characteristics and findings of each study, including the experimental model, lanthanum salt formulation, dosage, treatment duration, and primary outcomes, are detailed in Appendix A.

Exposure to lanthanum salts led to significant neural toxicity through multiple mechanisms, including the activation of apoptotic, autophagic, and inflammatory pathways, excessive ROS production, disruption of antioxidant defenses, mitochondrial dysfunction, and metabolic alterations (Table 1; Figure 3).

### 2.1. Clinical Findings of Lanthanum Toxicity

Studies reported in the literature concerning lanthanum-induced toxicity in rats focused on specific functions, including motor function, memory, as well as growth and development, with findings that were in some cases contradictory between research groups. To better emphasize these toxicological effects, each of these functions will be separately presented and discussed hereafter.

#### 2.1.1. Motor Function

Despite the cognitive impairments observed in rats exposed to lanthanum, their locomotion activity generally remained unaffected in most in vivo studies. Specifically, the rats exposed to LaCl_3_ did not have swimming speed and endurance that significantly differ from controls [34,50,51], with the exception of a study that reported impaired swimming endurance in Wistar rats exposed to a high dose of LaCl_3_ (40 mg/kg/day, representing milligrams of LaCl_3_ per kilogram of body weight per day) over a period of 5 months via oral gavage [52]. This was the longest exposure duration and the highest dosage among the studies.

Xiao et al. [27] reported impaired motor capacity of Sprague Dawley offspring rats exposed to a higher dose of lanthanum nitrate hexahydrate, noting decreased hindlimb strength, reduced running time and distance, and decreased grip strength. Notably, a different breed of rats and a distinct lanthanum compound were used.

In the nematode *Caenorhabditis elegans*, exposure to lanthanum nitrate hexahydrate resulted in significant reductions in locomotion abilities. L1 stage nematodes showed marked decreases in head thrash frequency, body bends, and pharyngeal pumping, with similar reductions also observed in L4 stage nematodes [35].

#### 2.1.2. Memory

In the Morris water maze test, a method used for assessing spatial memory and learning, offspring rats exposed to LaCl_3_ demonstrated significant, dose-dependent neurological deficits. The results revealed that LaCl_3_ exposure significantly impaired spatial learning and memory in a dose-dependent manner. Animals exposed to higher concentrations of LaCl_3_ exhibited longer escape latencies, swam longer distances, spent less time in the target quadrant, and were less likely to enter it compared to controls, with some failing to locate the platform. Their swimming patterns were more disorganized, characterized by chaotic and aimless movements, reflecting deficits in spatial navigation compared to controls [27,30,34,36,37,38,39,40,41,50,52].

Despite these cognitive impairments, motor function appeared to remain largely unaffected by LaCl_3_ exposure, as the observed increase in escape latency was not attributable to reduced exercise capacity, given that swimming speed and the ability to swim longer distances were comparable to those of the control groups. However, in two studies involving higher doses and prolonged exposure periods, mild impairments in motor abilities were noted [27,30].

Exposure of pregnant rats to LaCl_3_ also resulted in impaired avoidance-conditioned reflex abilities in their offspring, evidenced by a significantly longer electric shock duration and extended active avoidance latency, occurring in a dose-dependent manner [42].

Electrophysiological findings in Wistar offspring rats showed that LaCl_3_ significantly impacted long-term potentiation (LTP) in the hippocampus, a mechanism involved in learning and memory [37]. Following high-frequency stimulation (HFS), the population spike (PS) amplitude in rats treated with LaCl_3_ was significantly reduced compared to the control group, indicating that LaCl_3_ inhibited both the induction and maintenance of LTP, suggesting impaired hippocampal synaptic activity.

#### 2.1.3. Growth and Development

Exposure to lanthanum in rats has been associated with a range of adverse physiological effects, including dose-dependent inhibition of growth and development, as evidenced by decreased body and brain weights, impaired weight gain, and reduced food intake [27,30,35,38,40,42]. Conversely, Ding et al. reported no significant alterations in the weights of the whole brain, hippocampus, or cortex compared to control groups [42]. However, findings regarding hippocampal and brain coefficients are inconsistent: while some studies suggest that LaCl_3_ exposure in Wistar rats leads to elevated brain and hippocampus coefficients, others report no significant changes [38,40,42].

In summary, the behavioral findings from the included experimental studies reveal a distinct pattern of lanthanum-induced toxicity. The most uniform outcome is a significant, dose-dependent impairment of learning and memory, consistently observed across multiple studies. This contrasts with the effects on motor function and physical development, which are inconsistent and often contradictory. While some studies report deficits in locomotion or growth, these effects appear to be dependent on specific conditions, including high dosage, prolonged exposure, or the animal model used.

### 2.2. Toxicity Mechanisms

#### 2.2.1. Morphologic Damage

LaCl_3_ exposure has demonstrated significant neurotoxic effects, particularly in hippocampal regions and related neuronal structures, as seen in a range of in vivo and in vitro studies.

Exposure to LaCl_3_ resulted in significant neuronal damage, with important cytoplasmic alterations, including a marked reduction in organelles, vacuolated structures, an increased number of autophagosomes, swollen lysosomes and mitochondria, and an increased number of endosomes [40,42,43]. This mitochondrial damage was particularly evident in the CA1 region of the hippocampus and was correlated with a dose-dependent effect on the mitochondria-associated membrane (MAM) structures [34,39].

Reported nuclear alterations were severe, featuring pyknosis (nuclear condensation), smaller nucleoli, accumulation of scattered heterochromatin, and vacuolated structures within the nucleus and perinuclear spaces. Partial rupture of the nuclear membrane was observed after exposure to higher doses, highlighting the dose-dependent damage [40,43,44].

In the hippocampal CA1 region, exposure to LaCl_3_ caused significant axonal damage, characterized by a reduction in the number of axons, reduced length, and a twisted and clustered morphology, losing their typical linear structure, as highlighted with immunofluorescence staining with Tau-1, Map2, and DiL [45]. In the CA3 region of the hippocampus, a decrease in pyramidal neurons, reduced Nissl body expression, disorganized apical dendrites, and a reduced number of mossy fibers were reported [36,42,45].

In both the CA1 and CA3 regions, synaptic structures were disrupted, including uneven synaptic interfaces, shorter active zones, large intercellular gaps, thinner postsynaptic densities, fewer synaptic vesicles, and disordered neural cells, leading to impaired synaptic communication integrity [40,41].

Exposure to LaCl_3_ adversely impacted astrocytes, reducing their numbers and damaging their network connections, leading to shorter and fewer protrusions and resulting in incomplete structural integrity [53]. Additionally, an increase in glial cell infiltration in damaged hippocampal regions was described, indicating a neuroinflammatory component [46].

In *Caenorhabditis elegans*, exposure to lanthanum caused notable damage to dopaminergic and GABAergic neurons, including dendritic loss and soma damage. Higher concentrations of lanthanum resulted in increased α-synuclein aggregation in L1 stage nematodes, but not in the L4 stage nematodes [35].

#### 2.2.2. Axonal Growth

Exposure to LaCl_3_ resulted in abnormal axonal growth. Following exposure, key molecules in the LKB1 signaling pathway were downregulated, including LKB1, phosphorylated LKB1 (p-LKB1), STRAD, MO25, MARK2, α-tubulin, STK25, GM130, and significantly decreased ratios of p-MARK2/MARK2 and Tyr/Ace within the hippocampus were reported [45]. These suppressed pathways consequently affected the Golgi apparatus, resulting in microtubule post-translational modifications, mediated through the LKB1-STK25-GM130 and LKB1-MARK2 pathways [45].

#### 2.2.3. Blood–Brain Barrier (BBB) Disruption

Exposure to LaCl_3_ compromises BBB integrity in murine cerebral endothelial cells (bEnd.3) by increasing cytosolic calcium levels and activating the RhoA/ROCK signaling pathway. Jie Wu’s study [28] demonstrated elevated phosphorylated myosin light chain 2 (MLC2) levels and increased expression of endothelial MLCK protein, leading to actomyosin contraction and heightened BBB permeability, as indicated by reduced VE-cadherin expression and increased HRP permeation. Pretreatment with the calcium chelator BAPTA-AM mitigated these changes, implicating cytosolic calcium in the observed BBB damage.

#### 2.2.4. Synaptic Plasticity

Lanthanum chloride negatively affects the NF-κB signaling pathway, a critical mechanism for synaptic plasticity and memory. Reduced phosphorylation of IKKα/β and IκBα in LaCl_3_-treated rats prevented NF-κB activation, which suppressed the expression of immediate early genes (IEGs) such as *c-fos* and *c-jun* [31,41]. Additionally, brain-derived neurotrophic factor (BDNF) expression was significantly reduced within the hippocampus [41].

#### 2.2.5. Changes in Neurotransmitter Activity

Neurotransmitter systems are affected by lanthanum exposure in a dose-dependent manner. Reduced acetylcholinesterase (AChE) activity and increased acetylcholine (ACh) levels in the brain were reported, along with decreases in dopamine (DA), dihydroxyphenylacetic acid (DOPAC), norepinephrine (NE), and serotonin (5-HT) levels. Other authors reported that dopamine and serotonin plasma levels remained unchanged, while acetylcholine and norepinephrine levels were significantly decreased [27,30].

Following exposure, glutamate levels in the hippocampus were increased, while glutamine levels were reduced [37,43,44,46]. This disruption was due to decreased Na-K-ATPase activity and reduced expression of glutamate transporters (GLAST and GLT-1). Upregulation of mRNA and protein levels of NMDA receptor subunits NR1 and NR2B, but not NR2A, led to overactivation of NMDARs and subsequent neurodegeneration. These findings suggest that the excitotoxicity induced by LaCl_3_ is mediated through alterations in the glutamate–NO–cGMP signaling pathway, as also indicated by increased levels of nitric oxide (NO), cGMP, and increased iNOS activity in a dose-dependent manner [43,46].

### 2.3. Oxidative Stress as a Mechanism of Neuronal Toxicity

Exposure to LaCl_3_ induces a significant neurotoxic response in a dose-dependent manner, characterized by a marked increase in reactive oxygen species (ROS) production and disruption of antioxidant mechanisms (Figure 3) [34,44].

Following LaCl_3_ exposure, the nuclear factor erythroid 2-related factor 2 (Nrf2) pathway undergoes a marked downregulation at both the mRNA and protein levels, along with reduced expression of Nrf2-target genes, including Sod2, GSH-Px1, glutathione S-transferase (GST), and γ-glutamylcysteine synthetase (γ-Gcs-h) [40,46]. An impaired antioxidant response was suggested by the reduced mRNA levels of antioxidant enzymes, such as heme oxygenase-1 (HO-1) and NAD(P)H quinone dehydrogenase 1 (Nqo1). [34,47].

In a study performed on male CD-1 (ICR) mice, the activity of both enzymatic antioxidants—including SOD, catalase (CAT), and ascorbate peroxidase (APx)—and non-enzymatic oxidants, including ascorbic acid (AsA) and GSH, was depleted, and ROS production was increased [46]. These findings were corroborated in Wistar offspring rats: a dose-dependent upregulation of NADPH oxidase 4 (NOX-4) was observed, along with a reduction in GSH and SOD activity [34].

Regional differences were reported in studies of exposed Wistar rats, suggesting region-specific susceptibility to oxidative stress. In the hippocampus, CAT activity decreased significantly; SOD activity decreased primarily in the cerebral cortex, whereas glutathione peroxidase (GPx) activity decreased globally [36].

### 2.4. Apoptotic Pathways Involved in Neuronal Toxicity

#### 2.4.1. Calcium-Mediated Intrinsic Apoptosis

Exposure to LaCl_3_ led to elevated intracellular Ca^2+^ concentrations, particularly in primary cerebral cortical neurons [54]. This calcium overload was associated with the downregulation of anti-apoptotic proteins Bcl-2 and Bcl-xl and the upregulation of pro-apoptotic proteins Bax and Bad, resulting in an increased Bax/Bcl-2 ratio [38,54,55]. In both neurons and astrocytes, there was a dose-dependent reduction in pro-caspase-3 levels, with an increased expression of caspase-3 and caspase-9, while no significant change was observed in caspase-8 expression. These observations support the involvement of an intrinsic apoptotic mechanism mediated by mitochondria [44].

#### 2.4.2. Apoptotic Regulatory Pathway Suppression

LaCl_3_ exposure activates endoplasmic reticulum (ER) stress, as evidenced by increased expression of GRP78, GRP94, GADD153, phosphorylated JNK, and caspase-12 [44]. The PI3K/Akt/mTOR pathway was suppressed, with concomitant, dose-dependent reductions in PI3K, phosphorylated Akt (p-Akt), and mTOR expression, and decreased Akt mRNA levels [24,45,53]. Hypoxia-inducible factor 1-alpha (HIF-1α) and vascular endothelial growth factor (VEGF) were also diminished, suggesting impaired neurovascular signaling [42]. Downregulation of the NF-κB pathway, with reduced expression of phosphorylated IκBα, NF-κB p65, and brain-derived neurotrophic factor (BDNF), was reported, which plays a role in modulating neuronal plasticity and resistance to stress-induced damage [31]. Upregulation of miR-124 was associated with reduced phosphorylation of NF-κB p65 and reduced expression of postsynaptic density protein 95 (PSD-95) [41].

### 2.5. Inflammation

Exposure to lanthanum (III) chloride induces neuroinflammatory signaling by activating microglial cells with subsequent release of inflammatory mediators, in a dose-dependent manner (Figure 3). In vitro studies using BV2 microglia and primary cultured Kunming mouse cortical neurons demonstrated LaCl_3_-induced upregulation of ionized calcium-binding adapter molecule 1 (Iba1), along with the activation of the NF-κB signaling pathway, as indicated by increased phosphorylation of IKKα/β and IκBα and enhanced nuclear translocation of the NF-κB p65 subunit. The activation of microglia led to an increased production of nitric oxide (NO) and elevated expression of pro-inflammatory cytokines, including tumor necrosis factor-alpha (TNF-α), interleukin-1β (IL-1β), interleukin-6 (IL-6), and monocyte chemoattractant protein-1 (MCP-1) [48,50]. Exposure to LaCl_3_ in Kunming mice also resulted in microglial activation in the hippocampus, marked by increased Iba1 and elevated mRNA expression of TNF-α, IL-1β, IL-6, MCP-1, and inducible nitric oxide synthase (iNOS) [50].

Conditioned medium from LaCl_3_-treated microglia increased the proportion of apoptotic and necrotic neurons in co-culture models. This effect was attenuated by the NF-κB pathway inhibitor pyrrolidine dithiocarbamate (PDTC) [49].

Conversely, in vivo exposure of Wistar rats to LaCl_3_ resulted in downregulation of NF-κB signaling in the hippocampus, demonstrated by lower levels of phosphorylated IKKα/β and IκBα and reduced p65 activation [31].

### 2.6. Mitochondrial Toxicity

Lanthanum (III) chloride (LaCl_3_) exposure has been shown to cause extensive disruption of mitochondrial structure and function, characterized by compromised membrane integrity, impaired cellular metabolism, and activation of mitophagy (Figure 3).

In both primary neurons and astrocytes, LaC_l3_ induces a dose-dependent reduction in mitochondrial membrane potential, with decreased levels of mitochondrial cytochrome c and increased cytosolic cytochrome c, suggesting outer mitochondrial membrane permeabilization and activation of the intrinsic apoptotic cascade [24,49].

Ultrastructural analyses of affected neurons revealed significant morphological changes in the mitochondria, including swelling, vacuolization, cristae disruption, and blurred outer membranes, with the typical tubular mitochondrial morphology being replaced by degenerative, spherical forms, suggestive of severe structural damage [34,37].

Impairment of mitochondrial function led to metabolic consequences, with decreased ATP production, reduced cardiolipin content, suppressed expression of mitochondrial membrane protein TOM20, and reduced respiratory chain complex IV activity [37,39].

LaCl_3_-induced mitochondrial calcium overload was suggested by the increased expression of calcium transport regulators, including MCU, MICU1, and MICU2, which facilitate Ca^2+^ uptake into the mitochondrial matrix [39]. Mitochondrial dynamics were altered, with enhanced fission and suppressed fusion, as evidenced by increased levels of Drp1 and phosphorylated Drp1 at Ser616, decreased phosphorylated Drp1 at Ser637, and downregulation of mitofusins Mfn1 and Mfn2. Immunofluorescence analyses demonstrated increased colocalization of Drp1 with TOM20, supporting the translocation of fission machinery to the mitochondrial surface [34,39].

Excessive mitophagy was demonstrated by elevated levels of LC3B-II and mitochondrial-localized Parkin, along with decreased p62 expression. Although PINK1 transcription was unaltered, the protein accumulated on the outer mitochondrial membrane, facilitating Parkin recruitment [39].

Exposure to LaCl_3_ disrupts mitochondria-associated membranes (MAMs), as evidenced by a significant downregulation of several tethering complexes, including VAPB, PTPIP51, BAP31, FIS1, MFN2, and MFN1, along with the structural damage observed by electron microscopy [34].

### 2.7. Autophagy

Exposure of offspring Wistar rats to La_Cl3_ led to increased autophagic activity in the brain, particularly within the hippocampal regions CA1, CA3, and the dentate gyrus (DG) (Figure 3). A dose-dependent increase in autophagic flux was reported, reflected by the upregulation of autophagy-related proteins, including ULK1, Beclin1, ATG7, ATG12, LC3B-II, and Rab7, an increased LC3B-II/LC3B-I ratio, and decreased p62 expression [34]. LaCl_3_ has been shown to modulate downstream regulators of lysosomal fusion and autophagosome maturation, with increased expression of LAMP2 and STX17 and decreased levels of phosphorylated ULK1 and p62 [39]. Ultrastructural analysis confirmed the presence of autophagic vesicles and electron-dense vacuoles in neural cells [34].

### 2.8. Impacts of Lanthanum (La) on Energy Metabolism

Exposure to LaCl_3_ disrupts astrocytic glucose and lactate metabolism. In primary cortical astrocytes, LaCl_3_ induced a dose-dependent reduction in glucose transporter 1 (GLUT1), glycogen phosphorylase (GP), and glycogen synthase (GS) at both transcriptional and protein levels, with a decreased GS/p-GS ratio, indicating impaired glycogen synthesis and diminished astrocytic glycogen reserves. Monocarboxylate transporters MCT1, MCT2, and MCT4 were downregulated, indicating a reduced capacity for lactate transport from astrocytes to neurons [46].

Additionally, LaCl_3_ exposure altered lactate dehydrogenase (LDH) isoenzyme expression. Astrocytes showed decreased LDHA expression and higher LDHB expression, along with a decrease in total LDH activity [50]. In vivo findings from hippocampal tissue revealed a decrease in LDH activity, with downregulation of both GS and GP expression, indicating impaired glycolytic capacity [46]. These findings indicate impaired glycolytic activity and global energy metabolism, likely contributing to cognitive deficits in spatial learning and memory.

## 3. Discussion and Future Perspectives

The studies investigated for this review reveal that most of the research on REE toxicity is conducted in China. This geographic focus can be correlated with the high degree of exposure of its population to these elements due to the country’s dominant role in both the exploitation and production of REEs [6,17]. Because of its numerous processing facilities and acknowledged experience in the field, primarily owing to REEs’ natural abundance in this geographical area, other countries also deliver their extracted materials to China for further processing [56]. Consequently, numerous studies have documented higher concentrations of these emerging pollutants and associated health issues in populations living near Chinese mining, processing, and waste disposal sites [19,57]. While this is understandable, given China’s central role in REE mining and processing, this geographic concentration represents a significant limitation. It raises the possibility of regional publication bias and implies that the findings may not be generalizable to other populations with different genetic backgrounds, dietary habits, or environmental co-exposures. Research from other regions is critically needed to validate these findings globally.

Due to the accelerating demand for REEs [7], exposure levels are expected to rise considerably, thereby heightening the risk of adverse toxicological outcomes [58]. The increasing demand for REEs has prompted the diversification of extraction processes from unconventional sources, including freshwater streams, industrial residues, coal combustion byproducts, marine sediments, and electronic waste, among others, which underscores the significance of developing more economical and sustainable solutions [2].

Recycling REEs has been reported to present substantial opportunities for enhancing supply security, addressing the balance problem among REEs with varying levels of demand, and mitigating environmental risks associated with extraction and processing. Nevertheless, despite these benefits, safety concerns related to REE toxicity remain of high importance and necessitate further research to ensure safe implementation of recycling practices. In this regard, comprehensive and collaborative multidisciplinary approaches, combining technological innovation, environmental impact and health risk assessments, and economic viability evaluations, are essential to develop efficient practices for managing the growing challenges connected to REE exposure across their entire chain, from extraction to processing and recycling [58,59,60,61].

A limitation of the current work is that the described mechanistic insights are derived exclusively from cellular and animal models. Within these models, the evidence for lanthanum’s neurotoxicity is largely consistent, with dose-dependent findings of nervous system damage that are mechanistically supported by cellular studies. However, inconclusive data regarding specific cytotoxic mechanisms, such as the regulation of the NF-κB pathway, and conflicting clinical findings on growth, development, and motor function, warrant additional investigations. [27,30,31,35,38,42,50,51,52].

The evidence provides a plausible biological basis for adverse neurological outcomes following exposure to REEs, as similar observations have been made in human epidemiological studies [17,18,20]. The ability of REEs to cross the blood–brain barrier and accumulate within brain tissue in both rodents and humans, and the consistent reports of impaired learning and memory in exposed rodents, align with findings that environmental lanthanide exposure negatively influences cognitive functions in children [26,30,33,34,36,37,38,39,40,41,50,52].

Extrapolating these findings to humans must be done with caution, as several significant limitations must be acknowledged. Translational data can be biased by several factors, including differences in exposure dose and duration, when comparing the acute/subchronic focus of experimental studies to the low-dose, long-term environmental exposure in humans. Another factor is the route of administration, comparing the gastrointestinal intake of highly soluble lanthanum salts in studies to real-world exposure through contaminated water, food, and air pollution [2]. Finally, there is the limitation of using a single element in experiments, while environmental exposure involves mixtures of REEs [59].

The evidence synthesized in this review highlights several critical knowledge gaps and future research directions. Current data suggest an association between environmental exposure to REEs and neurodevelopmental disorders, such as neural tube defects and impaired cognition in children [17]. Considering the existing experimental data alongside the scarce epidemiological evidence, conducting cohort epidemiological studies, particularly in populations with high environmental exposure, could strengthen the link between REE exposure and adverse neurological outcomes. To better reflect real-world scenarios, future research should prioritize chronic, low-dose exposure in animal models to establish a more accurate dose–response relationship, especially as the longest chronic study to date was only five months [30,51].

## 4. Materials and Methods

A thorough literature search was performed up to July 2024 using the US National Library of Medicine, National Institutes of Health (PubMed), and Web of Science databases. The primary goal was to provide a mechanistic summary of existing data on the neurotoxicity of lanthanum salts in in vivo and in vitro experimental studies.

A broad search strategy was employed using a variety of keyword combinations, including “Rare Earth Elements”, “Rare Earth Metals”, “Lanthanide”, “Toxicity”, and “Neurotoxicity”. In PubMed, Medical Subject Headings (MeSH) terms were incorporated to enhance search precision. More specifically, the search query used was (“Metals, Rare Earth” [Mesh] AND (“toxicity” [Subheading] OR “Neurotoxicity Syndromes” [Mesh])), which yielded 1610 results after filtering for English-language articles with full-text availability. In the Web of Science database, the search equation used was: “Rare Earth Elements” AND (“Toxicity” OR “Neurotoxicity” OR “Central Nervous System”), which returned 1012 documents in English when the topic field tag (TS) was used. These included 881 articles, 111 review articles, 21 proceedings papers, 16 book chapters, and 17 additional documents belonging to other document types. Subsequently, the two datasets corresponding to the query results from PubMed and Web of Science were concatenated, and duplicate records identified by the “DOI” field were removed, resulting in a final set of 2336 documents. However, given the specificity of the investigated subject, a manual screening of the titles and abstracts was subsequently performed to identify the relevant articles for the present study.

The main inclusion criteria for this review comprised in vivo and in vitro studies on the toxicity of lanthanum(III) salts (LaCl_3_, La(NO_3_)_3_). Exclusion criteria included neurotoxicity studies of REEs other than lanthanum, review articles, inaccessible studies, and those published in languages other than English.

## 5. Conclusions

Exposure to REEs, particularly lanthanum salts, induces a complex, dose-dependent neurotoxic response characterized by the activation of apoptotic and inflammatory pathways, excessive ROS production, disruption of antioxidant defenses, and calcium signaling dysregulation. These mechanisms lead to neuronal injury and, ultimately, cognitive deficits, including learning and memory impairment.

Given the rising global use of REEs, these findings highlight a significant potential public health concern. Consequently, future efforts must prioritize establishing safe exposure limits through both experimental animal studies and large-scale human epidemiological research. Increased environmental monitoring of REEs in water and food supplies, particularly near industrial and e-waste sites, is advised. A deeper understanding of the long-term, low-dose effects of REE mixtures remains critical for developing effective public health policies and mitigating the risks posed by these emerging contaminants.

## Figures and Tables

**Figure 1 molecules-30-03748-f001:**
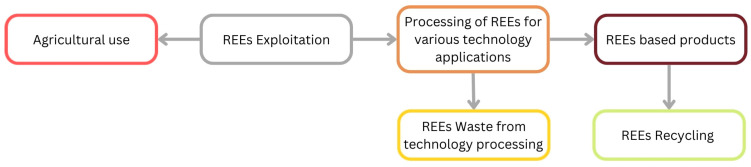
Main sources and transmission routes of REEs into the environment. This diagram illustrates the life cycle of REEs, from their initial exploitation and processing for various technological applications to their incorporation into products. The cycle highlights two key points of environmental release: waste generated during processing and the disposal or recycling of end-of-life products. The diagram is an original synthesis.

**Figure 2 molecules-30-03748-f002:**
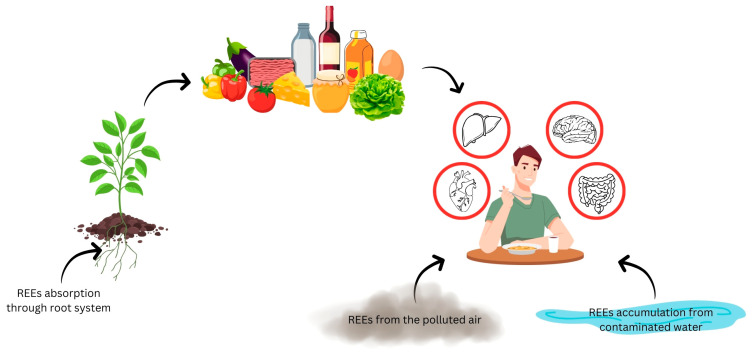
Primary pathways of REE exposure and their bioaccumulation in the human body. This diagram illustrates the exposure routes in humans, including consumption of contaminated food and water (originating from plant uptake from soil) and inhalation of polluted air [2]. Once absorbed, these elements accumulate in multiple organs, including the brain, liver, intestines, and heart. The diagram is an original synthesis.

**Figure 3 molecules-30-03748-f003:**
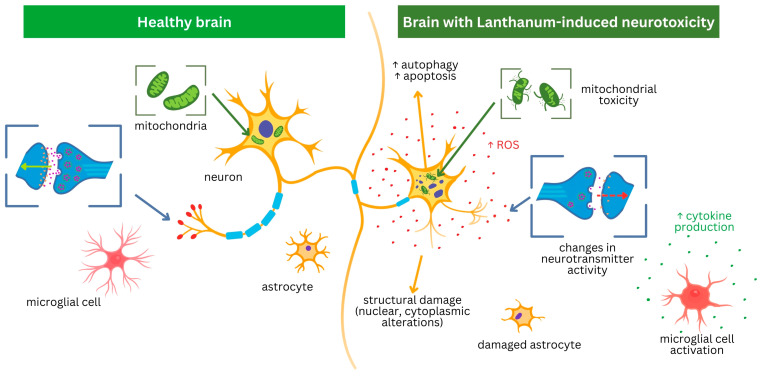
Key mechanisms of lanthanum (La)-induced neurotoxicity. The schematic contrasts a healthy brain environment (**left**) with the pathological changes following La exposure (**right**). Lanthanum causes direct neuronal injury through mitochondrial dysfunction, increased reactive oxygen species (ROS), and activation of cell death pathways (apoptosis, autophagy). Concurrently, it disrupts synaptic transmission and promotes neuroinflammation via astrocyte damage and microglial activation, leading to increased cytokine production. The figure is an original synthesis.

**Table 1 molecules-30-03748-t001:** Overview of the key mechanisms and cellular changes following lanthanum exposure.

Mechanism	Molecular/Cellular Changes	References
Cytotoxicity	↓ Cell viability; ↓ Astrocyte number; ↓ Cytoplasmic organelles; ↑ Vacuolated intracellular structures; ↑ Nuclear alterations; Impaired structural integrity; Impaired axonal growth; BBB disruption; Impaired synaptic plasticity; Disrupted neurotransmitter systems.	[28,31,34,35,36,37,38,39,40,41,42,43,44,45,46,47]
Oxidative Stress	↑ Prooxidants: ROS, MDA; ↓ Antioxidants: SOD, GSH, GPx, CAT, AsA; Nrf2 pathway downregulation.	[34,36,41,48]
Apoptosis	↑ Apoptotic rate and ↑ Bax/Bcl-2 ratio; ↑ Pro-apoptotic factors: Bax, Bad, Caspase-3, -9, -12; ↓ Anti-apoptotic factors: Bcl-2, Bcl-xl, pro-caspase-3; ↑ ER stress markers: GRP78, GRP94, GADD153; Suppression of PI3K/Akt/mTOR and NF-κB Pathway.	[24,31,38,42,44,48]
Inflammation	↑ Microglial activation (Iba1); ↑ Pro-inflammatory mediators (TNF-α, IL-1β, IL-6, MCP-1, NO, iNOS); Conflicting regulation of NF-κB pathway (in vitro vs. in vivo).	[30,48,49]
Mitochondrial Dysfunction	↓ Mitochondrial membrane potential; ↓ ATP production; ↑ Cytosolic cytochrome c release; ↓ Respiratory chain complex IV activity; Altered mitochondrial dynamics (↑ fission, ↓ fusion).	[24,34,37,39,48]
Autophagy	↑ Autophagosome formation; ↑ ULK1, Beclin1, LC3B-II; ↓ p62 (suggesting altered autophagic flux).	[34,39]
Energy Metabolism Alterations	↑ LDH release; ↓ Glucose and lactate transporters; ↓ GLUT1, ↓ MCT 1/2/4; ↓ Glycogen synthase (GS) & phosphorylase (GP).	[46,50]

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
