# Peer review of "Neurotoxicity of Lanthanum Salts: A Narrative Review of Mechanistic Insights from Cellular and Animal Models"

_molecules, 2025, doi:10.3390/molecules30183748_

Round 1
Reviewer 1 Report
Comments and Suggestions for Authors
The paper is interesting and well-structured; however I found some details that need to be corrected: the scientific names of Caenorhabditis elegans should be written in italics on lines 133 and 157. The names of the genes c-fos and c-jun should also be written in italics (line 251).
Author Response
Thank you for your valuable feedback. We have revised the manuscript according to your recommendations. Please see the attachment.

Reviewer 2 Report
Comments and Suggestions for Authors
The review describes the study of toxicity of lanthanum salts. Despite some strong points of the review, there are many questions and recommendations for it
The first paragraph provides very general information. This can be found in textbooks, lecture chapters, or even in an open electronic library like Wikipedia. References 1 and 2 are not suitable for what the authors described above: they should contain references to literary sources that describe the use of rare earth element compounds, which is what the authors describe. But only two references are provided about the biological activity of some rare earth element compounds and a review of possible future applications of rare earth elements.
Line 60: “the 60 harmful effects caused by these elements are no longer present only in countries such as China but have become widespread (Figure 1).” - With all due respect to the authors, figure 1 does not reflect what was written earlier in this sentence.
Line 63: “This is because the good mobility of LREEs ions facilitates their propagation and absorption from soil to plants, which subsequently represents the main entrance gate into the food chain (Figure 2)” - Figure 2 and the diagram applied to it - are there supporting references to literary sources with similar diagrams? I understand that everything looks logical, but I would still like to see several supporting scientific works, not reviews.
Line 81: “Based on the available data from the literature, the studies that analyze the effect that 81 these REEs have on human health report the use of Lanthanum salts – Lanthanum(III) chloride (LaCl3)” - Probably also at least a hexahydrate. Anhydrous salts of lanthanides are stored in sealed ampoules under an inert atmosphere.
Line 86 etc.: The authors started talking about lanthanum, specifying the object of the study, then again expanded the objects to all rare earth elements. I advise making the narrative more logical.
Lines 104-105 and 113-114 are most likely the same :
The present review is focused on the toxicity of lanthanum (La), as most toxicological studies have concentrated on this rare earth element, considered as providing an 'adequate' database for analysis [11]. This focus is also reflected in studies on neurotoxicity, with the most important proportion of studies using LaCl3
And
In the above-mentioned context, the aim of this work is to provide a comprehensive overview of the toxic and potentially beneficial effects of exposure to the most commonly studied REE — lanthanum (La) [11].
The authors have made a very correct observation. The fact is that gadolinium in various organic compounds is used as a contrast agent in magnetic resonance imaging, for example, in a drug such as "Gadobutrol". In this case, gadolinium is excreted from the body, clinical studies have been conducted.
In general, the authors say almost nothing about the use of rare earth elements as biologically active bioinorganic compounds - this must be added to the review. Because the availability of information about rare earth elements, economics and application is good, but the review is specifically about toxicity, this is worth concentrating on.
The authors use 28 articles where toxicity was studied on cells and mice. But at the same time, in figure 2, the authors draw a person, which is misleading. I have already written earlier that the same "Gadobutrol" is used everywhere in the world (+analogues) and the level of toxicity allows this to be done by repeatedly administering the drug.
Lines 146 etc: It is necessary to talk about the concentrations introduced to mice. And compare the concentrations in waste water, soil. Usually, all concentrations studied for toxicity on mice exceed the levels in the atmosphere by many orders of magnitude - this, on the contrary, can reduce the level of chemophobia, but not raise it.
Author Response

(The authors gave the same response as above.)

Reviewer 3 Report
Comments and Suggestions for Authors
This manuscript presents a narrative review of the neurotoxicity of lanthanum salts, focusing on mechanistic findings from cellular and animal investigations. The topic is highly relevant, given the increased environmental exposure to rare earth elements (REEs). The review is generally comprehensive, citing numerous mechanistic pathways, such as oxidative stress, apoptosis, mitochondrial damage, and synaptic impairment. The use of figures and tables helps enhance the ease of understanding the findings, while the discussion highlights the need for further studies.
Yet, there is room for improvement. The manuscript is repetitive at times, lacks discussion of the flaws of the cited research, and would be improved by more explicit organization of results and discussion. Problems with language and formatting also detract from readability. With revision, this paper has the potential to be a strong addition to the literature.
1. The review assembles a considerable amount of data, but novelty is moderate. The authors need to highlight how their synthesis goes beyond previous reviews on REE toxicity.
2. The majority of the manuscript is descriptive, recapitulating studies with inadequate critical assessment (e.g., discrepancies between animal models are stated but not discussed in depth). Please comment more on limitations and strength of evidence.
3. Almost all of the primary research was carried out in China. Although mentioned, the authors need to more critically comment on possible biases and generalizability of results.
4. The "Materials and Methods" section defines inclusion criteria, but information like the number of papers initially screened and how duplicates or low-quality studies were managed is lacking. A PRISMA-style flow diagram would be beneficial.
5. Table 1 and Figure 3 are useful but dense. You might summarize overlapping mechanisms (e.g., oxidative stress, apoptosis, mitochondrial dysfunction) in a more integrated conceptual scheme.
6. The majority of evidence comes from rodent models or in vitro experiments. Please include a section on translation to human exposure situations, with limitations of animal-to-human extrapolation.
7. Section 2.1 reports behavioral results, but interpretation is disjointed. A summary comparison (e.g., uniform findings in memory deficit, mixed findings in locomotor activity) would assist readers in seeing the patterns.
8. The diagrams are helpful but derived from common knowledge instead of being synthesized from the studies reviewed. Please cite sources and indicate if these are original diagrams.
9. The figure legends are too concise and not consistently self-descriptive. More complete, descriptive legends that define all abbreviations, symbols, and pathways should be provided by authors so that readers can comprehend the figures without having to refer to the main text again.
10. Some references are outdated or regionally limited. Consider including recent reviews on REE neurotoxicity (2022–2024) beyond lanthanum to broaden context.
11. Phrases like "beneficial effects" of lanthanum exposure are briefly noted but not defined. Please indicate if these are intended to refer to therapeutic applications (e.g., phosphate binding in CKD patients).
12. Where environmental routes are presented, the review could strengthen by quantifying the usual levels of exposure and contrasting them with experimental dosages, to emphasize ecological validity.
13. The discussion is mostly a reiteration of the results without further integration. Try to make this section stronger by suggesting research gaps (e.g., necessity for epidemiological research, dose-response relationship, chronic vs acute exposure differences).
14. The conclusion is brief but would more usefully emphasize actionable suggestions, e.g., recommendations for tracking environmental REEs or future directions for in vivo vs human cohort studies.
Comments on the Quality of English LanguageThe English needs moderate editing. Although the manuscript is readable, there are numerous grammatical errors, awkward sentences, and typos (e.g., "conducted to increased extraction" instead of "led to increased extraction"). A few sentences are too long or repetitive, decreasing clarity. Figure legends also do not provide enough detail and need to be rewritten more clearly.
Author Response

(The authors gave the same response as above.)

Round 2
Reviewer 2 Report
Comments and Suggestions for Authors
Dear Authors, Most of my recommendations were taken into account, which in my personal opinion improved the original manuscript
Good luck!
Author Response
Dear Reviewer,
Thank you for the constructive feedback!
Reviewer 3 Report
Comments and Suggestions for Authors
The revised version of the manuscript shows considerable improvement and a remarkable effort on the part of the authors to address all the concerns of the reviewers. The manuscript now provides a clearer account of the rationale for its originality, adds necessary discussions of limitations and bias, and enhances visual quality by more elegant figure legends and table layouts. The authors have done a more balanced integration of the results of the in vivo and the in vitro studies, recognized the geographical clustering of the dataset, and defined the requirements of future research. Furthermore, the careful consideration of the translational significance with respect to human exposure, albeit modest, supports the overall contribution.
Minor comments:
1. In the main text, it is crucial that every figure (e.g., Figure 3) is clearly referenced at the point where relevant mechanisms are being discussed. In particular, when explaining apoptosis or mitochondrial dysfunction, reference to Figure 3 would enhance the reader/observer's linking of the descriptive text to the pictorial representation.
2. In Section 2.1, define the unit usage of dosing descriptions (e.g., "40 mg/kg/day"). Be consistent throughout the paper and thought might be given to including human-relevant analogues where appropriate (even if rough estimates).
3. Although the authors detail why a PRISMA diagram could not be generated, a short table summarizing the number of articles that were screened/included by database (e.g., PubMed versus Web of Science) would improve transparency.
4. You may need to add a Supplementary Table of all the included studies with their model (animal or cell type), dose, duration, and outcome. It would be handy to the readers who would like to have an instant reference point without scrambling the table or text.
Author Response
Dear reviewer,
Please find attached our responses to your recommendations.
Thank you for all your efforts in enhancing the current work.
